Prognostic value of RNA methylation-related genes in gastric adenocarcinoma based on bioinformatics

He Xionghui 1
Chen Xiang 1
http://orcid.org/0000-0001-9966-0129 Yang Changcheng 2
Wang Wei 1
Sun Hening 1
Wang Junjie 1
Fu Jincheng 1
Dong Huaying 1 dr_dhy@163.com
1 Department of General Surgery, Hainan General Hospital, Hainan Affiliated Hospital of Hainan Medical University, Hainan Medical College , HaiNan, HaiKou , China
2 Department of Medical Oncology, The First Affiliated Hospital of Hainan Medical University, Hainan Medical College , HaiNan, HaiKou , China
Soares Paula
Electronic publication date: 2024 Feb 29
Publication date: 2024
Volume: 12
Electronic Location ID: e16951
Received 2023 Jul 6; Accepted 2024 Jan 24
Copyright: © 2024 He et al.
Copyright year: 2024
Copyright holder: He et al.
License: This is an open access article distributed under the terms of the Creative Commons Attribution License, which permits unrestricted use, distribution, reproduction and adaptation in any medium and for any purpose provided that it is properly attributed. For attribution, the original author(s), title, publication source (PeerJ) and either DOI or URL of the article must be cited.
License URL: https://creativecommons.org/licenses/by/4.0/

Keywords: Stomach adenocarcinoma, RNA methylation, Risk model, Immune microenvironment, Bioinformatics

Funding: The Key Research and Development Program of Hainan province ZDYF2021SHFZ055 Hainan Provincial Natural Science Foundation of China 822CXTD535 National Natural Science Foundation of China 81960475 This work is supported by The Key Research and Development Program of Hainan province (ZDYF2021SHFZ055), the Hainan Provincial Natural Science Foundation of China (822CXTD535) and the National Natural Science Foundation of China (81960475). The funders had no role in study design, data collection and analysis, decision to publish, or preparation of the manuscript.

==============================
Background

Gastric cancer (GC) is a malignant tumor that originates from the epithelium of the gastric mucosa and has a poor prognosis. Stomach adenocarcinoma (STAD) covers 95% of total gastric cancer. This study aimed to identify the prognostic value of RNA methylation-related genes in gastric cancer.

Methods

In this study, The Cancer Genome Atlas (TCGA)-STAD and GSE84426 cohorts were downloaded from public databases. Patients were classified by consistent cluster analysis based on prognosis-related differentially expressed RNA methylation genes Prognostic genes were obtained by differential expression, univariate Cox and least absolute shrinkage and selection operator (LASSO) analyses. The prognostic model was established and validated in the training set, test set and validation set respectively. Independent prognostic analysis was implemented. Finally, the expression of prognostic genes was affirmed by reverse transcription quantitative PCR (RT-qPCR).

Results

In total, four prognostic genes (ACTA2, SAPCD2, PDK4 and APOD) related to RNA methylation were identified and enrolled into the risk signature. The STAD patients were divided into high- and low-risk groups based on the medium value of the risk score, and patients in the high-risk group had a poor prognosis. In addition, the RNA methylation-relevant risk signature was validated in the test and validation sets, and was authenticated as a reliable independent prognostic predictor. The nomogram was constructed based on the independent predictors to predict the 1/3/5-year survival probability of STAD patients. The gene set enrichment analysis (GSEA) result suggested that the poor prognosis in the high-risk subgroup may be related to immune-related pathways. Finally, the experimental results indicated that the expression trends of RNA methylation-relevant prognostic genes in gastric cancer cells were in agreement with the result of bioinformatics.

Conclusion

Our study established a novel RNA methylation-related risk signature for STAD, which was of considerable significance for improving prognosis of STAD patients and offering theoretical support for clinical therapy.

Introduction

One of the most common malignancies in the world is gastric cancer (GC). According to statistics, more than 1 million individuals are diagnosed with stomach cancer each year, accounting for 5.7 percent of all cancer diagnoses, and around 700,000 people die from it each year (Ye et al., 2021; Wong et al., 2021). GC, which is produced by malignant alterations in gastric gland cells and accounts for 90 percent to 95 percent of GC (Zhang et al., 2020), is the most frequent pathological form of GC. Despite the fact that the prognosis for GC has improved as a result of the advancement of numerous therapeutic options, including as surgery, chemotherapy, radiation and targeted therapy, the patients’ 5-year overall survival rate is still dismal (Qiu et al., 2020). As a result, it is critical to thoroughly comprehend GC’s molecular process and to investigate novel and dependable prognostic indicators.

In RNA, more than 160 chemical changes have been found (Zhang et al., 2021), and the m6A (N6 methyladenine) change in eukaryotic mRNA and lncRNA was detected as early as the 1970s, showing that methylation of RNA is a post-translational regulation mechanism (Zhou et al., 2020). The most frequent methylation modification of mRNA in eukaryotic cells is m6A, which has an impact on all phases of the RNA life cycle (Roundtree et al., 2017a). M6A RNA affects splicing (Tang et al., 2018; Yao et al., 2021), export (Roundtree et al., 2017b), stability and translation of mRNA posttranscriptionally (Huang et al., 2018). Five-methylcytosine (m5C) is found in the mRNA transcript’s untranslated region (UTR), and it has been linked to a variety of gene expression activities such as RNA export, ribosome assembly and translation (Pan, Huang & Xu, 2021). N1-methyladenosine (m1A), a critical posttranscriptional alteration in RNA, was discovered more than five decades ago and has now acquired general acceptance (Zhao et al., 2019). The majority of m1A is found at the translation initiation site of mRNA, which is close to the translation start point of mRNA (Gao et al., 2021). M7G controls mRNA export, translation and splicing by being located at the caps of mRNAs, as well as at particular internal sites in tRNAs and rRNAs (Pandolfini et al., 2019; Zhang et al., 2019).

RNA methylation-related proteins include methyltransferase (Writer), demethylase (Eraser), and RNA methylation-specific recognition protein (Reader) (Zhou et al., 2020). RNA methylation-related proteins have been shown to reversibly govern essential biological activities as RNA metabolism, processing and stem cell directed differentiation (Zhao et al., 2014). In recent years, RNA methylation has been linked to tumor genesis, progression and metastasis. For example, According to one study, DNMT1-mediated FOXO3a promoter hypermethylation reduces FOXO3a expression in breast cancer, and FOXO3a decreases breast cancer stem cell characteristics and tumorigenicity through lowering FOXM1/SOX2 signaling (Liu et al., 2020). The RNA-binding protein ALYREF binds to the m5C gene. PKM2 and ALYREF levels have been associated to poor outcomes in bladder cancer patients, suggesting that ALYREF and its target gene PKM2 might be helpful biomarkers for guiding early bladder cancer diagnosis (Wang et al., 2021). WDR4 levels are linked to cancer immunity and may be used as a prognostic biomarker for some cancers (Zeng et al., 2021). However, RNA methylation in GC is infrequently examined, and the overall prediction level in GC is insufficient.

We investigated the expression of RNA methylation regulators in STAD using the The Cancer Genome Atlas (TCGA) and Gene Expression Omnibus (GEO) datasets, and for the first time looked at the relationship between RNA methylation-related prognostic genes and clinicopathological features. To demonstrate the predictive usefulness of prognostic genes related with RNA methylation for Stomach adenocarcinoma (STAD), we created several tumor subgroup models and risk models. It was also highlighted how these differentially expressed genes relate to cell signaling pathways and the tumor immunological microenvironment.

Materials and Methods

Data source

This study retrived the TCGA-STAD cohort from the TCGA database (https://portal.gdc.cancer.gov/). The TCGA-STAD cohort contained 32 normal tissue samples and 375 STAD tissue samples, of which 345 STAD samples containing intact clinical information were included in the cluster analysis, prognostic analysis and correlation analysis with clinical characteristics. The GSE84426 dataset (Yoon et al., 2020), which comprised 76 GC samples bearing survival time for the validation of the prognostic model, was excavated from the GEO database (https://www.ncbi.nlm.nih.gov/). RNA methylation genes (RMGs) were obtained according to previous reports (Sun et al., 2020; Xie et al., 2020; Huang et al., 2021) and were enumerated in Table 1.

Table 1 RNA methylated genes enrolled in this study.

RNA methylation	Writer	Reader	Eraser	
M1A	TRMT10C		ALKBH1	
	TRMT61B		ALKBH3	
	TRMT6			
	TRMT61A			
M5C	NSUN1 (NOP2)	ALYREF	TET2	
	NSUN2			
	NSUN3			
	NSUN4			
	NSUN5			
	NSUN6			
	NSUN7			
	DNMT1			
	DNMT2 (TRDMT1)			
	DNMT3A			
	DNMT3B			
M6A	METTL3	HNRNPC	ALKBH5	
	METTL14	YTHDC1	FTO	
	RBM15	YTHDC2		
	WTAP	YTHDF1		
	ZC3H13	YTHDF2		
	KIAA1429			
M7G	METTL1			
	WDR4			

Discrepant expression analysis

The differentially expressed genes (DEGs) between control and STAD specimens in the TCGA-STAD cohort were defined utilizing the ‘limma’ R package (Smyth, 2004). The cut-off criterion was |log2FoldChange(FC)| > 0.5 and p value < 0.05. The volcano map and heatmap were produced using the ‘ggplot2’ and ‘pheatmap’ R package. The differentially expressed RNA methylation genes (DERMGs) were obtained by crossing DEGs and RNA methylation genes (RMGs).

Consensus clustering

DERMGs associated with prognosis were selected by univariate Cox analysis. A total of 345 STAD cases in the TCGA-STAD cohort were classified depending on the expression of these genes applying the ‘ConsensusClusterPlus’ R package for consensus clustering (Wilkerson & Hayes, 2010). The clustering results was verified by t-SNE dimension reduction analysis.

Functional annotation analysis

Gene Ontology (GO) and the Kyoto Encyclopedia of Genes and Genomes (KEGG) enrichment analysis was execute through ‘clusterProfiler’ R package (Yu et al., 2012). The p value < 0.05 and count ≥ 2 represented significance. GO analysis was comprised of biological processes (BP), cellular components (CC) and molecular functions (MF) correlated with genes.

Establishment of RNA methylation-relevant risk signature in STAD

The 345 STAD patients in the TCGA dataset were separated randomly into a training set (242 cases) and a test set (103 cases) at a 7:3 ratio. Through univariate Cox analysis and least absolute shrinkage and selection operator (LASSO) logistic regression in the training set, the RNA methylation-relevant genes significantly associated with prognosis were filtered out. Using the formula: Riskscore = ∑1n⁡coef(genei)∗expr(genei) and median of RNA methylation-relevant risk score, patients were separated into two risk subgroups (high risk and low risk). GSE84426 dataset was served as external validation set. Kaplan-Meier (K-M) curves (‘survminer’ package), receiver operating characteristic (ROC) analysis (‘survivalROC’ package), and risk curves (‘ggplot2’ package) were deployed to prove the predictive efficiency of RNA methylation-relevant risk signature.

Independent prognostic evaluation and generation of a nomogram model

The relevance between RNA methylation-relevant risk signature and clinicopathological factors, containing age, gender, T stage, N stage, M stage and pathological tumor stage was detected by Wilcoxon test or Kruskal-Wallis test and shown by violin plot. K-M analysis of the designated subtypes with different clinicopathological factors was also carried out. Cox regression analyses (Univariate and multivariate) were executed to recognize independent predictors of OS in the TCGA-STAD dataset. After that, a nomogram comprising the independent predictors of overall survival (OS) was created via the ‘cph’ function in R. The calibration curves and ROC curves (1-, 3- and 5-year) were drawn to affirm the predictive accuracy of the nomogram.

Gene set enrichment analysis

By using the log2(FC) of genes between two RNA methylation-relevant risk subgroups as the ranking criterion, Gene Set Enrichment Analysis (GSEA) was implemented by ‘clusterProfiler’ R package on the basis of GO and KEGG gene sets. The threshold for significantly enriched items and pathways was adjusted p value < 0.05.

The relevance analysis between RNA methylation-relevant prognostic genes and TME

The Tumor Immune Single Cell Hub (TISCH) (http://tisch.comp-genomics.org/) (Sun et al., 2021) is a database focused on the TME, collecting 76 tumor datasets from 27 cancers and involving single-cell transcriptomic profiles of almost 2 million cells. We used the GSE134520 dataset from the TISCH database to recognize the type and distribution of cells in TME, and the expression of RNA methylation-relevant prognostic genes in TME-associated cells.

The correlation analysis between prognostic genes and immune-modulator

TISIDB (http://cis.hku.hk/TISIDB/) is a comprehensive repository portal of tumor-immune system interactions (Ru et al., 2019) and was employed to analyze the Spearman correlations between RNA methylation-relevant prognostic genes and immune modulators.

The mutation analysis of prognostic genes

The mutation of prognostic genes in the STAD samples of TCGA-STAD cohort were analyzed in cBioPortal (https://www.cbioportal.org/) (Unberath et al., 2019). The association of copy number variants (CNV) with the expression of prognostic genes was evaluated by chi-squared test.

RNA isolation and RT-qPCR

Firstly, the one normal gastric epithelial cell line (GES-1) and three GC cell lines (MKN-45, MKN-7 and SNU-1) was fully lysed by the TRIzol Reagent according to the producer’s directions (Ambion, Austin, TX, USA). Chloroform (Chengdu Guerda rubber industry Co., LTD, Chengdu, China) was used to remove proteins and fat-soluble magazines, ice isopropanol (Chengdu Guerda rubber industry Co., LTD, Chengdu, China) was used to precipitate RNA, and 75% ethanol (Chengdu Colon Chemical Co., LTD, Chengdu, China) was used to further remove impurities. Then, the RNA concentration was detected with NanoPhotometer N50. After that, the reverse transcription of RNA into cDNA was done via the SweScript-First-strand-cDNA-synthesis-kit (Servicebio, Wuhan, China) relying on the manufacturers’ protocol (The four cell lines are all derived from iCell Bioscience Inc., a biotechnology company based in Shanghai, China). The qPCR reaction system was displayed in Table 2. The reactions were implemented on a CFX96 real-time quantitative fluorescence PCR instrument. The amplification reactions were detailed in Table 3. The primer sequences for PCR were tabulated in Table 4. The expression was uniformized to the internal reference GAPDH and computed employing the 2−ΔΔCq method (Livak & Schmittgen, 2001).

Table 2 The reaction system of qRT-PCR validation.

Component	Volume	
cDNA	3 ul	
2×Universal Blue SYBR Green qPCR Master Mix	5 ul	
Forward primer (10 µM)	1 ul	
Reverse primer (10 µM)	1 ul	

Table 3 The amplification procedures of PCR system.

	Temperature	Time	
Pre-denaturation	95 °C	1 min	
Denaturation	95 °C	20 s	
Annealing	55 °C	20 s	
Extendsion	72 °C	30 s	

Table 4 The sequences of the primers for qPCR.

Symbol	Sequence	
ACTA2 For	GCATCCTCATCCTCCCTT	
ACTA2 Rev	CTCCTTCTGCATTCGGTC	
SAPCD2 For	TGACCGAGAAGAGTGAGC	
SAPCD2 Rev	TGAAGGTGGAATCCAGAG	
PDK4 For	GTGATGTGGTAGCAGTGGTC	
PDK4 Rev	TCTATTGGTGTAAGGGAAGG	
APOD For	ACTAATGGAAAACGGAAAGA	
APOD Rev	GTACAGGAATACACGAGGGC	
GAPDH For	CCCATCACCATCTTCCAGG	
GAPDH Rev	CATCACGCCACAGTTTCCC	

Statistical analysis

All bioinformatic study were conducted using the R language, and the data from different groups were compared by the Wilcoxon test or Kruskal-Wallis test. Significance of differences in RT-qPCR was determined by t-test. P values less than 0.05 were deemed to be statistically significant if not noted above.

Results

DERMGs in STAD

The detailed flowchart is shown in Fig. S1. In the TCGA-STAD dataset, a total of 4,937 DEGs (STAD vs normal), containing 3,647 up-regulated genes and 1,290 down-regulated genes in STAD, were uncovered (Table S1, Fig. 1A). Hence, 14 DERMGs (NOP2, DNMT1, YTHDF1, NSUN2, TRMT6, METTL1, WDR4, RBM15, KIAA1429, NSUN5, DNMT3B, TRMT10C, ALYREF and ZC3H13) between STAD and normal samples were determined by crossing the DEGs and RMGs (Fig. 1B). The expression of DERMGs association with M1A and M5C modify, including TRMT10C, TRMT6, NOP2, NSUN2, NSUN5, DNMT1, DNMT3B, ALYREF, was shown in Figs. 1C and 1D. Meanwhile, the expression of DERMGs association with m6A and M7G modify, including RBM15, ZC3H13, KIAA1429, YTHDF1, METTL1 and WDR4, was revealed in Figs. 1E and 1F. All the 14 DERMGs presented a upward expression trend in STAD samples in comparison to normal samples. The GO enrichment analysis revealed that methylation, RNA modification, and tRNA-related pathways were notably enriched in these 14 DERMGs, such as RNA methylation, tRNA modification, mRNA (Fig. 1G).

Figure 1 Differential RNA methylation regulatory gene screening.

(A) Volcano plot of DEGs between STAD and normal samples. (B) The intersection of DEGs and RNA methylation regulation genes. (C) The violin map of the expression of DERMGs association with M1A and M5C modify. (D) The expression heat map of DERMGs. (E) The violin map of the expression of DERMGs association with m6A and M7G modify. (F) The expression heat map of DERMGs association with m6A and M7G modify. (G) The GO terms enriched in DERMGs. Four asterisks (****) represent p < 0.0001.

Recognition of DERMGs-related subtypes in STAD

The above 14 DERMGs were then subjected into the univariate Cox analysis and four genes associated with overall survival (OS) of STAD patient (p < 0.05) were identified, namely RBM15, DNMT1, WDR4 and ALYREF (Fig. 2A). To probe the related subtypes in STAD, 345 STAD samples were clustered based on the expression of four RMGs using consensus cluster analysis. From the result, two subtypes were identified, designated as cluster 1 and cluster 2 (Figs. 2B–2E). The cluster 1 contained 164 cases and cluster 2 contained 181 cases. The t-SNE result revealed that cluster 1 and cluster 2 could be clearly separated, further demonstrating the reliability of the clustering result (Fig. 2F). Survival analysis revealed that prognosis differed significantly between the two DERMGs-related subtypes, and cluster 1 had a notable survival advantage (Fig. 2G). The expression of RBM15, DNMT1, WDR4 and ALYREF were up-regulated in cluster 1 (Fig. 2H). Since the hazard ratio (HR) of RBM15, DNMT1, WDR4 and ALYREF was all less than 1 (Fig. 2A), above results indicated that RBM15, DNMT1, WDR4 and ALYREF were protective factors in influencing the survival of STAD patients.

Figure 2 Identification of DERMGs-related subtypes in STAD.

(A) Forest plot of univariate Cox of four DERMGs associated with overall survival (OS) in STAD patients. (B) Relative change in area under CDF area for k = 2 to 6. (C) The cumulative distribution functions. (CDF) for k = 2 to 6. (D) Tracking plot for k = 2 to 6. (E) Consensus score matrix of all samples when k = 2. (F) t-SNE dimension reduction analysis verifies the reliability of the clustering results. (G) Kaplan-Meier survival curve based on two clusters. (H) The expression heat map of four DERMGs associated with OS of STAD patients in two clusters and different clinicopathological subgroups.

Following this, we deployed the ‘limma’ package to mine the DEGs between cluster 1 and cluster 2 utilizing p value < 0.05 and |Log2FC| > 1 as screening criteria. As shown in Figs. S2A, S2B and Table S2, 56 DEGs (10 up-regulated genes and 46 down-regulated genes) between cluster 1 and cluster 2 were mined. We performed functional enrichment analysis on the 56 genes, and a grand total of 40 GO items (12 BP items, 21 CC items and seven MF items) and 1 KEGG pathway were derived (Table S3). The top eight items in each classification were exhibited in Figs. S2C and S2D. We noted that the DEGs were associated with digestion, maintenance of gastrointestinal epithelium and mitosis-related biological processes.

The RNA methylation-relevant risk signature based on DEGs between cluster 1 and cluster 2

To dig the genes relevant to the overall survival (OS) of STAD patients, we incorporated the 56 DEGs between cluster 1 and cluster 2 into a univariate Cox analysis in the training set. Five out of the 56 genes were certificated as genes linked to patients’ OS in the training set (p-value < 0.05) (Fig. 3A). Subsequently, the five genes were further submitted to LASSO regression analysis. Four feature genes (ACTA2, SAPCD2, PDK4 and APOD) were picked out and each regression coefficient was computed (Figs. 3B and 3C). We then constructed a risk score model with the following formula: Riskscore = 0.279 × ACTA2 + (−0.16759) × SAPCD2 + 0.087985 × PDK4 + 0.0665 × APOD. Based on this formula, we calculated the risk score for each STAD patient in the training set (cutoff = 0.763) and test set (cutoff-0.705) and classified them into high- and low-risk subgroups based on median value respectively. K-M curves manifested that patients with higher risk had noteworthy worse survival than patients with lower risk (Figs. 3D and 3E). The AUC values OS in the training cohort were 0.625 (1-year), 0.604 (3-year) and 0.703 (5-year) respectively, reflecting a decent accuracy of the model (Fig. 3G). Meanwhile, the AUC values of OS in the test set were 0.721 (1-year), 0.60 (3-year) and 0.683 (5-year) respectively (Fig. 3H). Survival status manifested that as the RNA methylation-relevant risk score increases, patients suffered a relatively high risk of death (Figs. 3J and 3K). The expression heatmap manifested APOD, ACTA2 and PDK4 were highly expressed in the patients with higher risk. SAPCD2 were highly expressed in the patients with lower risk (Figs. 3J and 3K). To further confirm the reliability of the RNA methylation-relevant risk signature, the above analysis was carried out in the external validation set (GSE84426, cutoff = 3.848). The results of external validation set were consistent with the training set (Figs. 3F, 3I and 3L).

Figure 3 Construction and validation of prognostic risk profiles in STAD patients with four survival related genes.

(A) Forest plot of univariate Cox of five genes associated with OS of STAD patients. (B and C) Four prognostic genes were selected using LASSO regression. (D–F) Kaplan-Meier survival curve of high- and low-risk group of training set, testing set, and validation set. (G–I) The 1, 3, and 5 years ROC curve verifies the specificity and sensitivity of risk feature prediction in training set, testing set, and validation set. (J–L) The risk curve in training set, testing set, and validation set according to the risk score. The risk curve is composed of the upper, middle and lower parts, and the abscissa is the patient samples sorted according to the risk score. In (J–L), the upper figure is consistent with the abscissa in the middle figure, and the risk score increases successively from left to right. The ordinate is risk score and time to survival, and the dotted line is the median risk score and the corresponding number of patients. In the (J–L) figures, the lower figure is a heat map of model gene expression in the high- and low-risk group.

Risk score and clinical features

Next, we further explored the relationship between the RNA methylation-relevant risk score and clinical characteristic factors by comparing risk scores across clinical characteristic subgroups and stratified survival analysis. As exhibited in Fig. 4, patients aged less than 60 years had significantly higher risk than those aged greater than 60 years. Stage I patients had significantly lower risk than those in Stage II–IV. The result of stratified survival analysis indicated that the risk score was an effective predictor of survival in N1–N3 stage patients, M0 stage patients, T3–T4 stage patients, stage III–IV patients, male patients and patients aged greater than 60 (p value < 0.05, Fig. S3).

Figure 4 Association between clinicopathological features and prognostic risk scores.

Differences of risk scores in clinical characteristics among different groups, including age (A), gender (B), pathologic M stage (C), pathologic N stage (D), pathologic stage (E) and pathologic T stage (F).

Tumor microenvironment analysis between the two RNA methylation-relevant risk subgroups

We conducted difference analysis on immune cells, molecular drugs and SNP mutations in the two RNA methylation-relevant risk subgroups of the training set. Tumor-infiltrating immune cells with CIBERSORT showed that naive B cells, memory B cells, activated CD4 memory T cells, follicular helper T cells, resting NK cells, monocytes, macrophages M0, resting mast cells, and activated mast cells had significant distinctions between the two risk subgroups (Figs. S4A and S4B). SNV waterfall plot showed that altered in 105 (89.74%) of 117 samples in high-risk subgroup (Fig. S4C), and altered in 114 (95.8%) of 119 samples in a low-risk group (Fig. S4D). The correlations between risk score and TMB between two risk groups showed significant differences (Figs. S4E and S4F). The drug sensitivity between the two risk groups was shown in Fig. S4G, 36 drug sensitivities were notably different in between the two risk subgroups (Table S4). Finally, we analyzed the survival difference among TMB high-risk high group, TMB high-risk low group, TMB low-risk high group and TMB low-risk low group, the results showed significant difference (Fig. S4H).

Independent prognostic value of the RNA methylation-relevant risk signature

The result of Cox analysis (univariate) demonstrated that the p values for stage, T stage, N stage, M stage and risk score were all less than 0.05 (Fig. 5A) and the corresponding factors were enrolled in the multivariate Cox analysis. The forest plot of Cox regression analyses (multivariate) demonstrated that the risk score and stage were independent predictors of prognosis for patients with STAD (Fig. 5B). The nomogram containing independent prognostic predictors was generated (Fig. 5C). The C index of the nomogram was 0.671, and the calibration plots manifested that the nomogram was an accurate predictor (Fig. 5D).

Figure 5 Independent prognostic analysis.

(A) The forest plot of univariate Cox analysis to select the features related to patients’ survival. (B) The forest plot of multivariate Cox analysis to select the independent prognostic factors. (C) The nomogram includes independent prognostic factors to predict the survival of patients. (D) The 1, 3, and 5 years of calibration curve of nomogram.

GSEA analysis

To probe possible reasons for the inconsistency in survival between the two RNA methylation-relevant subgroups, we proceeded with a GSEA analysis. A total of 2,658 GO entries and 132 KEGG pathways were derived (Table S5). Top 10 GO entries and KEGG pathways were shown in Figs. 6A and 6B. We noted that biological processes such as ‘mitotic sister chromatid segregation’, ‘cell cycle checkpoint’, ‘DNA damage checkpoint’, ‘G2/M transition of mitotic cell cycle nuclear division’, ‘RNA splicing, via transesterification reactions’, ‘RNA splicing, via transesterification reactions with bulged adenosine as nucleophile’ were linked to the low-risk subgroup (Fig. 6A). Meanwhile, ‘pathways in cancer’, ‘Olfactory transduction’, ‘Neuroactive ligand-receptor interaction’, ‘PI3K-Akt signaling pathway’, ‘MAPK signaling pathway’, ‘Cytokine-cytokine receptor interaction’, ‘Calcium signaling pathway’, ‘Ras signaling pathway’, ‘Proteoglycans in cancer’ and ‘Focal adhesion’ were correlated with the high-risk subgroup (Fig. 6B). We speculated that cancer- and immune-related pathways played an vital role in influencing the prognosis of the high-risk subgroup.

Figure 6 GSEA results for the high- and low-risk groups.

(A) Top 10 GO entries in GSEA analysis between high- and low-risk groups. (B) Top 10 KEGG pathways in GSEA analysis between high- and low-risk groups.

Prognostic genes and TME

As TME plays a role in gastric carcinogenesis and progression, we analyzed the expression of RNA methylation-relevant prognostic genes in TME-associated cells using the GSE134520 dataset in the TISCH database. As shown in the Figs. 7A and 7B, we analyzed nine cell clusters in the GSE134520 dataset, and Figs. 7A and 7B showed the number and distribution of the nine cell clusters. Further analysis revealed that ACTA2 was mainly expressed in myofibroblasts and fibroblasts (Figs. 7C and 7G), APOD had the highest expression in fibroblasts (Figs. 7D and 7H), PDK4 was mainly expressed in fibroblasts, myofibroblasts and malignant (Figs. 7E and 7I), and SAPCD2 was mainly expressed in pit mucous and gland mucous (Figs. 7F and 7J).

Figure 7 Correlation between prognostic genes and TME in STAD and expression of biomarkers in different cell clusters.

(A and B) The cell types and their distribution in the GSE134520 dataset. (C–F) Using the TISCH database, the correlation between the expression of APOD, ACTA2, PDK4 and SAPCD2 and TME in STAD tissues was analyzed. (G–J) The distribution of APOD, ACTA2, PDK4 and SAPCD2 in different cell types was analyzed using single-cell resolution in the GSE134520 dataset. STAD, gastric adenocarcinoma; TME, tumor microenvironment; TISCH, tumor immune single-cell center.

Prognostic genes and immune factor

The immune system exerts an important role in influencing both gastric carcinogenesis and progression as well as patient prognosis. Therefore, we analyzed the correlation between RNA methylation-relevant prognostic genes and immune factors. The results showed that ACTA2 was significantly positively correlated with the immunoinhibitors CSF1R, PDCD1LG2, TGFB1, TGFBR1 and immunostimulators CD48, CXCL12, CXCR4, ENTPD1, TMEM173, TNFRSF13B, TNFSF4 (Figs. S5A–S5K, S6A and S7A). APOD was significantly positively related to the immunoinhibitors ADORA2A, CSF1R, PDK4 and the immunostimulators TGFBR1, C10orf54, CXCL12, CXCR4, ENTPD1, TNFRSF13B (Figs. S5L–S5S, S6B and S7B). PDK4 was significantly positively associated with immunoinhibitors TGFBR1, and immunostimulators C10orf54, CXCL12, CXCR4, ENTPD1, TNFRSF13B (Figs. S8A–S8F, S6C and S7C). SAPCD2 was significantly negatively related to the immunoinhibitors BTLA, CSF1R and immunostimulators C10orf54, CD28, CD40LG, CD48, CXCR4, IL6R (Figs. S8G–S8O, S6D and S7D). SAPCD2 and the immunostimulator PVR was significantly positively correlated (Figs. S8P and S7D).

Mutation of prognostic genes in STAD

The cBioPortal website was used to analyze the mutation status of the four prognostic genes in the STAD sample. Four prognostic genes all had missense mutations and amplification, and ACTA2 and APOD had deletions (Fig. 8A). The specific mutation sites for each gene were shown in Fig. 8B. We further analyzed the effect of CNV on the expression of RNA methylation-relevant prognostic genes and detected that the expression of RNA methylation-relevant prognostic genes was significantly associated with CNV (Fig. 8C).

Figure 8 The mutation analysis of prognostic geness.

(A) Summary of alterations in prognostic genes in STAD through cBioPortal. (B) The distribution of mutation sites of prognostic genes. (C) Correlation between the prognostic genes and copy number variation.

The interaction and correlation analysis of prognostic genes

To investigate the interaction among prognostic genes, the protein-protein interaction (PPI) network including four prognostic genes and 20 genes that interact with them was constructed via GeneMANIA database. Figure S9A indicated all four prognostic genes interacted strongly with 20 genes, especially with APOA2, GC, and MYL12A. In addition, PDK4, PDP1, and PDRP were mainly involved in functions such as regulation of sulfur metabolic process, regulation of nucleotide biosynthetic process, and acetyl-CoA metabolic process. In addition, we explored the correlation of four prognostic genes and 14 DERMGs. The results showed that PDK4, APOD and ACTA2 were negatively correlated with DERMGs, while SAPCD2 was positively correlated, except for ZC3H13 (Fig. S9B).

Validation of prognostic gene expression

As illustrated in Table S1, SAPCD2 was up-regulated and ACTA2, PDK4 and APOD were presented reduced expression in STAD tissues compared to control tissues. To further affirm the expression of prognostic genes, we collected one normal gastric epithelial cell line (GES-1) and three GC cell lines (MKN-45, MKN-7 and SNU-1), extracted RNA, and executed RT-qPCR. As exhibited in Fig. S10, we detected that SAPCD2 was highly expressed in GC cell lines, while ACTA2, PDK4 and APOD were lowly expressed in GC cell lines. The experimental result proved that the expression trends of prognostic genes were comparable to with the result from public database. Moreover, we test the methylation levels of the three feature genes and found they differed in methylation sites, ACTA2 detected one differentially methylated site, APOD had a total of two methylated sites, only one of which differed, and PDK4 had two differentially methylated sites (Fig. S11).

Discussion

GC is the major cause of mortality from neoplastic disorders (Wu et al., 2019), with gastric adenocarcinoma accounting for about 95% of stomach cancers (Liu et al., 2017). As has been shown in recent decades, the development of stomach cancer is a complex process involving several components, and different molecular manifestations have variable clinical prognostic repercussions for patients. Tumor staging in clinical practice only evaluates tumor size, histological classification and metastasis (Niimi et al., 2013), but not the patient’s gene expression level. Recent research has demonstrated that stomach cancer development is influenced by epigenetic changes (Zhang et al., 2018). Non-coding RNA, histone modification, RNA methylation and DNA methylation are all examples of epigenetics (Chen et al., 2016; Xie et al., 2021), with RNA methylation becoming a popular topic in recent years. It is a dynamic and reversible modification mechanism, with methyltransferases, demethylases and binding proteins regulating the whole process (Mo et al., 2020; Romanowska et al., 2021). Evidence is accumulating in GC, underscoring the influence of RNA methylation on the malignancy and prognosis of tumor cells. For example, A recent investigation has suggested that METTL3 enhances the proliferation, formation of colonies, migration, and invasion of GC cells in an m6A-dependent manner via the pre-B-cell leukemia homeobox 1 (PBX1)/GTP cyclohydrolase 1 (GCH1)/tetrahydrobiopterin (BH4) axis (Liu et al., 2022).

By downloading and retrieving a large number of GC samples from the TCGA database, we observed that the expression of 14 RNA methylation regulators was significantly altered in GC tissues. NOP2, DNMT1, YTHDF1, NSUN2, TRMT6, METTL1, WDR4, RBM15, KIAA1429, NSUN5, DNMT3B, TRMT10C, ALYREF and ZC3H13 are among the genes that have been identified. All 14 DERMGs were found to be elevated. According to our GO enrichment analysis, these DERMGs are significantly associated with RNA methylation, tRNA modification, and related processes. Previous research suggests that the expression of NOP2 primarily relies on the m5C methylation level, as it can promote cancer cell proliferation through m5C-dependent inhibition of Cyclin-Dependent Kinase Inhibitor 1B (CDKN1B, p27Kip1) (Mei et al., 2020; Yang et al., 2023). The m7G tRNA methylation facilitated by the METTL1/WDR4 complex selectively enhances the translation of specific cyclin and oncogenic transcripts, along with downstream pathway-related mRNAs, thus governing cell proliferation and apoptosis in conjunction with the abundant homologous codons of m7G tRNAs (Porta, Paglino & Mosca, 2014). Hence, we speculation that the elevated expression of these DERMGs may exert a significant impact on their target transcripts, especially those involved in cancer development, rendering them pivotal epigenetic regulatory factors. This mechanism could potentially serve as a common oncogenic driver in GC progression.

Following that, we used univariate COX analysis to find four genes, RBM15, DNMT1, WDR4 and ALYREF, that were linked with overall survival (OS) in STAD patients. RBM15, DNMT1, WDR4 and ALYREF were identified as protective factors affecting STAD patient survival. Previous research has demonstrated that silences DNMT1 gene activity and leads to the activation of a number of tumor-related target genes (Chen et al., 2018). WDR4 is a methyltransferase that participates in cell cycle progression, signaling, gene control and apoptosis (Michaud et al., 2000; Arrondel et al., 2019). WDR4 gene expression was found to be down-regulated in advanced renal papillary carcinoma but constant in advanced rectal adenocarcinoma and GC in one study (Zeng et al., 2021). WDR4 expression was also found to be a protective factor in rectal adenocarcinoma in the corresponding OS outcome study. ALYREF is thought to be a preventive factor against colon cancer (Huang et al., 2022). This is in line with the findings of our research. RBM15 may have a function as an oncogene in lung adenocarcinoma, according to relevant publications (Li et al., 2020); Su, Huang & Hu (2019) discovered that RBM15 may contribute to the malignant evolution of GC and alter clinical prognosis. This might be due to the fact that RBM15 influences the formation of malignant tumors via a variety of molecular processes.

Furthermore, we discovered that the RNA methylation genes APOD, ACTA2, PDK4 and SAPCD2 are linked to the prognosis of GC. Simultaneously, these four genes have been shown to reliably predict patient prognosis in predictive risk profiles. Patients who meet the criteria for being classified as high-risk have a bad prognosis. These data show that RNA methylation-associated prognostic genes may have predictive prognostic value as well, and as a result, they might be employed as a potential prognostic predictor in patients with stomach cancer.

APOD, ACTA2, PDK4 and SAPCD2 have all been linked to the development of cancer in several studies. APOD is a protein that is generated in many different regions of the body (Braesch-Andersen et al., 2014). APOD-related complexes deliver cholesteryl esters to the liver and encourage their breakdown (Utermann et al., 1980). Additionally, apolipoproteins in the blood may provide lipids to cancer cells, giving them energy for expansion and invasion (Xu et al., 2007). Compared to women without breast cysts, those with breast cysts had a threefold higher chance of developing BC (Zhou & Luo, 2020). Because APOD is the most abundant component of cyst fluid, it is possible that APOD plays a role in the development of breast cancer (Sánchez et al., 1992). APOD was substantially expressed in the high-risk group, increased in colon cancer tissues, and had potential predictive value, according to a correlation study in the TCGA database against colon cancer (Liang, Su & Wu, 2021). Hu et al. (2018) discovered that APOD expression was substantially connected with the diagnosis of GC as a risk factor, and that it might be a possible prognostic biomarker for GC. ACTA2 promotes cell mechanical tension and shape maintenance, as well as tumor cell metastasis (Lambrechts, Van Troys & Ampe, 2004). Patients with high ACTA2 expression in lung adenocarcinoma had considerably increased distant metastases and a poor prognosis, according to one research (Lee et al., 2013). ACTA2 may hasten the formation of human tumors by acetylating late 6-phosphogluconate dehydrogenase (6PGD) at K76 and K294 (Shan et al., 2014). ACTA2 inhibition causes an intracellular buildup of unesterified oxysterols, which limits the development of hepatoma cell lines and xenografts (Lu et al., 2013). He et al. (2022) discovered that greater ACTA2 expression was related with a worse clinical outcome in the GC high-risk group. PDK4 belongs to a group of isozymes (PDK1-4) that help the body convert to aerobic glycolysis by transferring pyruvate metabolism from the mitochondria to the cytoplasm for glycolysis (Woolbright et al., 2018). Because most cancers have some degree of hypoxia, Hypoxia-inducible factor 1 (HIF-1) is connected to a poor prognosis in bladder cancer and is assumed to be the key mediator of transcriptional control of aerobic glycolysis in cancer (Chai et al., 2008; Denko, 2008), could be the main transcription factor driving PDK4 upregulation in cancer. PDK4 is a biomarker of GC prognosis and a prospective therapeutic target, according to Liu, Zhang & Suo (2021). It is also linked to the quantity of invading immune cells and patients’ poor prognosis, and it is a biomarker of GC prognosis as well as a viable therapeutic target in GC. PDK4 may have predictive relevance in various malignancies, such as liver and colorectal cancer (Leclerc et al., 2017; Deng et al., 2018; Qin et al., 2020). Furthermore, we discovered for the first time, based on the existing prognostic model, that SAPCD2 expression was connected with survival in STAD patients and was recognized as a protective prognostic factor. SAPCD2 is a recently discovered highly conserved mammalian gene that regulates cell cycle and chromosomal segregation (Mao et al., 2014). This gene is substantially expressed in GC, glioma, liver cancer and other tumor tissues, according to relevant research, and is linked to tumor malignancy, invasion and metastasis (Wan et al., 2014; Ni et al., 2015; Jayanthi, Das & Saxena, 2020). We think that SAPCD2’s role in the formation of STAD is complicated and dynamic, but the molecular mechanism has to be investigated further. Overall, Our study suggests that the RNA methylation-related risk signatures we constructed may serve as prognostic indicators for clinical treatment decisions in GC patients.

Several investigations have shown that genetic alterations often result in phenotypic abnormalities linked to carcinogenesis (Binang et al., 2020; Chen et al., 2021). CNV changes have been shown to disrupt genes involved in cell cycle regulation, the complement system and antigen presentation, potentially contributing to cancer (Brown et al., 2016). APOD and ACTA2 were deleted, and four prognostic genes, APOD, ACTA2, PDK4 and SAPCD2, were mutated and substantially related to CNV, according to our analysis. Following this study’s GSEA analysis, certain biological mechanisms have been found to have an important part in the onset and progression of cancer. Chromosome instability is a critical diagnostic sign for most malignancies (Olson et al., 2010); it is caused by abnormal cell division during mitosis or by changes in gene expression levels following chromosomal separation. Defects that guide proper and accurate chromosomal separation during mitosis can cause chromosome aneuploidy and instability, which can lead to malignant tumor formation and spread (Zhu et al., 2015). A cell cycle checkpoint is a crucial quality control measure that ensures that cell cycle events proceed normally. Cell growth will become uncontrollable if the regulatory method fails (Yin et al., 2018). Loss of checkpoint control of cell cycle pathways occurs with cancer growth, and their dysregulation may impact the efficiency of cancer treatment (Zhang et al., 2011; Ho et al., 2017). Furthermore, the quick transition of the G2/M phase during mitosis promotes GC cell growth (Lim et al., 2020). By controlling cell cycle progression and tumor growth, the PI3K/AKT signaling pathway increases the onset and progression of GC (Pereira et al., 2020). Many studies have shown that the MAPK/ERK signaling pathway is the primary molecular mechanism behind liver carcinogenesis (Zender et al., 2010), and that HCC can be efficiently generated by activating oncogenes or inhibiting tumor suppressor genes in conjunction with MAPK/ERK signaling (Zhang, Budker & Wolff, 1999). The inflammatory response has been linked to cytokine–cytokine receptor interactions, and cancer and inflammation have a very strong connection (Kan et al., 2019). Pro-tumor inflammation accelerates the growth of tumor cells and, to some extent, reduces anti-tumor immunity (Yu, Pardoll & Jove, 2009).

TME, which includes tumors, extracellular matrix, lymph, blood vessels and interstitial cells, plays a crucial part in the development, invasion and metastasis of tumors (Palicelli et al., 2021). According to one research, myofibroblasts are the most common stromal cell type in most malignancies, and they have been proven to aid tumor development by secreting a range of growth factors (Sharma, Evans & Hemers, 2016). At the same time, myofibroblasts can produce a variety of pro-tumor factors and directly participate in promoting metastasis (Karnoub et al., 2007; Seiler et al., 2020). Hwang et al. (2008) discovered that pancreatic tumor fibroblasts have a key role in tumor cell proliferation, motility, invasion and treatment resistance. By stimulating epithelial cell transition into mesenchymal cells and secreting energy-rich substances to enhance cancer cell development, fibroblasts boost prostate cancer aggressiveness. Fibroblasts also promote capillary morphogenesis and endothelial cell proliferation (Taddei et al., 2014). Disruptions in the quantity of mucous cells in the antrum and homeostasis have been reported to increase the probability of GC (Yu et al., 2016). Glandular mucus cells release glandular mucin, help make gastric mucin, and subsequently form a mucus gel layer to protect the stomach mucosa from the outside world (Ota & Katsuyama, 1992). At the same time, glandular mucin secreted by glandular mucus cells can inhibit helicobacter pylori infection to prevent GC, and can also be used as an inhibitor of differentiated gastric adenocarcinoma (Nakayama, 2014). In this study, we found that ACTA2, APOD and PDK4 were mainly expressed in myofibroblasts and fibroblasts respectively, while SAPCD2 was mainly expressed in alveolar mucous cells and glandular mucous cells of gastric adenocarcinoma. We speculated that ACTA2, APOD and PDK4 were risk factors. It may promote the onset and progression of STAD via the aforementioned mechanism, while SAPCD2 plays a protective role.

In recent years, immune infiltration has been a research focus. PDK4 expression was shown to be favorably linked with the numbers of several invading immune cells, including CD4+T cells, B cells and dendritic cells, as well as macrophages (Chai et al., 2008). APOD was shown to be a risk factor and substantially linked with lymphoid infiltration in a biologic information investigation on colon cancer (Liang, Su & Wu, 2021). Many immunosuppressive agents, such as csF1R, ADORA2A, TGFBR1 and BTLA, were discovered to be related with APOD, ACTA2, PDK4 and SAPCD2 in this research. In addition to immunosuppressive agents, we discovered immune stimulators such as CD48, CXCR4, C10 and PVR. APOD, ACTA2, PDK4 and SAPCD2 are thought to be essential in the immunomodulation of STAD. As a result, combining inhibitors and activators might possibly improve STAD patients’ anti-cancer effects.

Pharmacological chemotherapy is still often used to treat GC patients, but the development of chemoresistance during treatment is a key factor in patient fatalities. Clinically, patients with different tumor stages often respond differently to various chemotherapy treatments. We carried out chemotherapy sensitivity analysis for high and low risk categories so that GC patients might have superior antitumor chemotherapy regimens. Dasatinib, one of the most extensively studied tyrosine kinase inhibitors in clinical research, was initially developed as a dual inhibitor targeting BCR-ABL and SRC (Li et al., 2010). In other cancers, Dasatinib is capable of impeding cell migration and invasion through the inhibition of SRC, which blocks the transmission of downstream signals to proteins like focal adhesion kinase (FAK), responsible for adhesion, as well as MAPK and p27, which are involved in regulating the cell cycle (Johnson et al., 2005; Nam et al., 2005). In GC, dasatinib leads to a notable rise in actin accumulation at the cell cortex, which is also indicative of an enhanced cell-cell adhesion (Fenton, Hutchens & Denning, 2015; Montenegro et al., 2020). BIBW2992 not only directly inhibits HER2 receptor activation in GC but also suppresses its downstream signaling pathways, such as PI3K/AKT/mTOR and MAPK (Keller et al., 2018). Additionally, BIBW2992 induces apoptosis and arrests the cell cycle at the G1 phase (Chen et al., 2019). However, the effectiveness and applicability of these drugs still need validation through clinical trials.

We summarized four RNA methylation regulators, divided them into two subclasses according to their expression, and then analyzed them through the study of RNA methylation. We believe that more characteristic genes are often not conducive to large-scale clinical application (Li et al., 2021). Therefore, the four prognostic genes related to RNA methylation that we analyzed will be conducive to the successful implementation of clinical practice. These prognostic genes express differently in the high-risk and low-risk groups, which indicates that they are important in the incidence rate and progression of STAD. We established a risk model to study the clinical relevance of prognostic genes associated with RNA methylation. The AUC value and ROC curve show that compared with previous studies (Guan et al., 2020), the risk model we built has more advantages in predicting patient survival. In addition, this study found that the expression of prognostic genes is related to patient survival, which is an independent factor in the prognosis of STAD. At the same time, we also found that stage is an independent predictor, and we created a nomogram as a clear reference. It is worth noting that compared with the previous study (Guan et al., 2020; Wang, Meng & Ma, 2021), we increased the use of real-time PCR experiments to verify the expression of characteristic genes. In conclusion, this is the first time to study the predictive significance of RNA methylation related disease subtypes and RNA methylation related genes in STAD. These genes were linked to STAD prognosis and have many research prospects and may be good biological targets.

While our findings are promising, we must acknowledge that our research has significant limitations. First, bigger samples are required in clinical investigations to confirm the clinical importance of the built characteristics as independent prognostic markers of STAD, which will be the subject of future study. Second, while we validated the differential expression of prognostic genes in clinical samples through qRT-PCR, further investigations involving the establishment of animal or cell models will facilitate our understanding of the mechanisms underlying these prognostic genes in STAD. Additionally, the findings in our study, including the applicability of the prognostic model and the efficacy of drugs with varying sensitivities, require validation in clinical samples to demonstrate their clinical relevance. These areas represent the focal points of our future research endeavors.

Supplemental Information

Supplemental Information 1 The detailed flowchart.

Supplemental Information 2 Identification of genes related to RNA methylation in STAD.

(A) The volcano map of DEGs between culster2 and culster1. (B) The heatmap of DEGs between culster2 and culster1. (C) The top 8 BP entries, top 8 CC entries, and 7 MF entries were enriched by DEGs between culster2 and culster1. (D) The KEGG pathways were enriched by DEGs between cluster2 and cluster1.

Supplemental Information 3 Stratified survival analysis based on different clinicopathological features.

Supplemental Information 4 Association of ACTA2, PDK4, APOD and SAPCD2 with immunosuppressive factors in all cancers.

Supplemental Information 5 Correlation between ACTA2 (A-K), APOD (L-S) and immunomodulators.

Supplemental Information 6 Heatmap of the correlation between immunoinhibitors and RNA methylation-related prognostic genes in multiple cancers.

(A) Correlation between ACTA2 and immunoinhibitors. (B) Correlation between APOD and immunoinhibitors. (C) Correlation between APOD and immunoinhibitors. (D) Correlation between SAPCD2 and immunoinhibitors.

Supplemental Information 7 Heatmap of the correlation between immunostimulators and RNA methylation-related prognostic genes in multiple cancers.

(A) Correlation between ACTA2 and immunostimulators. (B) Correlation between APOD and immunostimulators. (C) Correlation between APOD and immunostimulators. (D) Correlation between SAPCD2 and immunostimulators.

Supplemental Information 8 Correlation between PDK4 (A-F), SAPCD2 (G-P) and immunomodulators.

Supplemental Information 9 Analysis of prognostic genes.

(A) The PPI network of prognostic genes. (B) The relevance of prognostic genes and DERMGs. **p < 0.01.

Supplemental Information 10 The expression of prognostic genes in cell lines detected by RT-qPCR.

(A) ACTA2 (B) SAPCD2 (C) PDK4 (D) APOD *p < 0.05, **p < 0.01, ***p < 0.001, ****p < 0.0001.

Supplemental Information 11 Differentially methylated site of three feature genes.

Supplemental Information 12 DEGs between STAD and normal samples in the TCGA cohort.

Supplemental Information 13 56 DEGs in Cluster 1.

Supplemental Information 14 GO Entries and KEGG Paths Functionally Enriched by 56 DEGs in Cluster 1.

Supplemental Information 15 36 drug susceptibilities in high-risk and low-risk groups.

Supplemental Information 16 Functionally Enriched GO Entries and KEGG Paths in GSEA.

Supplemental Information 17 Clinical data of TCGA in GC.

Supplemental Information 18 GSE84426.

Supplemental Information 19 Raw code.

Supplemental Information 20 test.coxinput.

Supplemental Information 21 train.coxinput.

Supplemental Information 22 Raw data: End Point Results.

Supplemental Information 23 Raw data: Quantification Summary.

Supplemental Information 24 Raw data: Quantification Plate View Results.

Supplemental Information 25 Raw data: Quantification Cq Results.

Supplemental Information 26 Raw data: Quantification Amplification Results.

Supplemental Information 27 Raw data: Melt Curve Summary.

Supplemental Information 28 Raw data: Melt Curve RFU Results.

Supplemental Information 29 Raw data: Melt Curve Plate View Results.

Supplemental Information 30 Raw data: Melt Curve Peak Results.

Supplemental Information 31 Raw data: Melt Curve Derivative Results.

Supplemental Information 32 Raw data: Standard Curve Results.

Additional Information and Declarations

Competing Interests

Author Contributions

Data Availability

The authors declare that they have no competing interests.

Xionghui He conceived and designed the experiments, analyzed the data, prepared figures and/or tables, and approved the final draft.

Xiang Chen conceived and designed the experiments, analyzed the data, prepared figures and/or tables, and approved the final draft.

Changcheng Yang conceived and designed the experiments, analyzed the data, prepared figures and/or tables, and approved the final draft.

Wei Wang performed the experiments, authored or reviewed drafts of the article, and approved the final draft.

Hening Sun performed the experiments, authored or reviewed drafts of the article, and approved the final draft.

Junjie Wang performed the experiments, authored or reviewed drafts of the article, and approved the final draft.

Jincheng Fu performed the experiments, authored or reviewed drafts of the article, and approved the final draft.

Huaying Dong conceived and designed the experiments, analyzed the data, prepared figures and/or tables, and approved the final draft.

The following information was supplied regarding data availability:

The raw data and the code are available in the Supplemental Files and from GEO (GSE84426 and GSE134520) and GDC (https://portal.gdc.cancer.gov, TCGA-STAD).

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
