# Peer review of "Prognostic value of RNA methylation-related genes in gastric adenocarcinoma based on bioinformatics"

_PeerJ, doi:10.7717/peerj.16951_

## Round 0.1 · original submission · Major Revisions

Overall, the reviewers agree that this is a well-written article that adds value to the field. However, there are several major concerns related to methodologies, data processing, interpretation of results, and significance. Please review and address all the comments and suggestions from the reviewers.

Reviewer 1 ·

Basic reporting

General Assessment:
The manuscript attempts to establish the prognostic value of RNA methylation-related genes in stomach adenocarcinoma (STAD). The manuscript is comprehensive and employs multiple datasets and bioinformatics tools to identify key genes that could serve as independent prognostic indicators. However, there are several areas that could be improved for clarity, rigor, and impact.

Specific Comments:

The abstract provides a good overview but could benefit from a succinct summary of the main findings, particularly the clinical implications.

The methodology regarding RT-qPCR could be expanded for clarity, including a detailed description of the sample preparation and RNA isolation.

Were any corrections made for multiple testing? This is crucial when dealing with high-dimensional data.
Like, line 234, i believe the correction, such as bonferroni correction, are necessary. The same for Figure 7 C.

Experimental design

No comment.

Validity of the findings

No comment.

Additional comments

No comment.

Reviewer 2 ·

Basic reporting

The authors aimed to investigate the prognostic value of RNA methylation-related genes in gastric cancer. Ultimately, they identified four genes (APOD, ACTA2, PDK4, SAPCD2) and constructed a risk prognostic model for assessing the prognosis of gastric cancer. Additionally, they explored the correlation between these genes and tumor immune microenvironment and mutations. Overall, the article has a relatively well-structured and clear presentation, although there are concerns that need to be appropriately addressed prior to publication.

My concern is listed below:
1. In the flowchart of Supplementary Figure 1, the mention of SVM-RFE refers to the Support Vector Machine Recursive Feature Elimination algorithm, correct? If so, it appears that this algorithm was not utilized in either your article or the provided source code. Please let me know if it was used and kindly point it out. Additionally, in your article, you mentioned using the TCGA dataset to obtain DEGs and RMGs genes. However, the dataset displayed in your flowchart is GSE36059, which pertains to kidney transplantation data. This flowchart seems unrelated to the structure of your article.

2. In line 187, the original text indicates the identification of 4 upregulated genes and 50 downregulated genes. However, upon reviewing your supplementary files, it seems that the actual numbers are 10 upregulated genes and 46 downregulated genes. Please carefully review your data.

3. In line 198, there are only five genes mentioned here. Does it still make sense to perform LASSO regression with such a small number of genes? As I am not an expert in statistics, I cannot determine its feasibility. If it is reasonable, please provide the rationale or supporting evidence.

4. In lines 202 to 204, the original text mentions dividing the training and testing sets into high-risk and low-risk subgroups based on their respective medians. This approach is generally not reasonable because you constructed the model and obtained the risk scores' median from the training set. To validate your model using the validation and external datasets, it would be more appropriate to use the risk score's median from the training set to divide the validation and external datasets into groups. This approach will better reflect the accuracy of the model. Additionally, I noticed in your source code that you set a random seed when performing LASSO regression. Can you provide the median risk score in the article?

5. The prognosis analysis of APOD, ACTA2, and PDK indicates that they are considered risk factors. However, the RT-qPCR results show low expression in cancer tissues. What could be the reason behind this discrepancy? Have you compared the differential expression of these genes between cancer and adjacent tissues in a database? Does it align with your experimental findings?




Minor suggestions/recommendations:
1. In scientific writing, gene names are typically presented in italics. Please refer to similar articles published by this journal for guidance on the expression of gene names.

2. In scientific writing, it is customary to provide the full name or explanation of proper nouns that are mentioned for the first time in the abstract, rather than using abbreviations. For example, in line 6 , "TCGA",and in line 9, "LASSO"

3. In line 44, the expression of "M6A" in the article may not comply with standard conventions. Typically, when "M6A" appears at the beginning of a sentence, it is capitalized, while in other cases, it is usually represented as "N6-methyladenosine (m6A)." Please refer to similar articles published by this journal for guidance on modifying the expression accordingly. You can check out https://peerj.com/articles/16023/ for example.


4. When utilizing datasets from the GEO database in the Materials and Methods section of the article, it is important to cite the corresponding publications for the data obtained from other researchers.

5. In line 107, does the sentence "Via univariate Cox analysis LASSO logistic regression in the training set" lack an "and"? Is the expression 'LASSO logistic' correct here? I have reviewed the source code you provided, and it seems that the 'glmnet' function uses the 'Cox' parameter.

6. In line 187, there might be an error in the expression. It should be "in cluster 1 and cluster 2" instead of "in cluster1"

7. In line 210, there is an error in the expression. It should be the ACTA2 gene instead of ACAT2. Additionally, ACAT2 appears multiple times throughout the text. Please make the necessary corrections.

8. In line 291, there seems to be an error in the result description. The majority of APOD genes appear to have amplification.

9. There is an error in line 234. "TME" should be corrected to "TMB".

10. In line 264, there appears to be an error in the description of the dataset. GSE14520 is a liver dataset, which does not align with the figure. Please make the necessary correction.

11. The language in the abstract can be slightly refined to enhance fluency and readability.

Experimental design

no comment

Validity of the findings

no comment

Reviewer 3 ·

Basic reporting

The article is written professionally and presents a comprehensive study on the role of RNA methylation in the development and progression of gastric cancer (GC), integrating data from various recent studies and the TCGA database. The text provides a reasonable background on the existing literature regarding the role of epigenetic changes, specifically RNA methylation, in the development of stomach cancer. The authors effectively underline the importance of looking beyond traditional tumor staging parameters to include investigations into gene expression levels. It is encouraging to see an exploration of the specific RNA methylation regulators that are notably altered in GC tissues, pointing towards a promising direction in GC research. While the authors succeed in shedding light on this complex topic, offering novel insights into the potential prognostic markers and therapeutic targets for GC, the text, however, suffers from a certain lack of depth in the discussion and critical analysis of the findings.

Some of the figure labels are too small to read and the figure quality is not allowing to zoom in like in fig 1a, 2f, 2h, 3d-f, 5e-f etc.

Experimental design

While the text succeeds in identifying potential protective factors affecting STAD patient survival, it seems to treat each gene in isolation, not offering a synthesized view on how these genes might interact or be interconnected in the larger network of RNA methylation processes that influence GC. A more detailed analysis of how these genes and their associated processes might function in synergy or opposition could provide a more nuanced view of the GC molecular landscape.

Moreover, the section that identifies the 14 RNA methylation regulators seems to be a pivotal point in the text, yet it somewhat fails to fully explore the implications of these findings. The statement that these regulators were found to be elevated is somewhat hanging, requiring further explanation and analysis. What could be the potential repercussions of this elevation in the context of GC? How do these findings integrate with the existing body of research? A more detailed discussion would provide readers with a fuller understanding of the importance of these discoveries.

Validity of the findings

The review appears somewhat limited in its critical analysis of the cited studies and its own findings. The text references a range of studies highlighting the roles of different genes and their links to various cancers, including GC, but stops short of delving deep into the mechanisms through which these genes exert their effects. While it is recognized that the METTL3, YTHDF1, and other genes play a crucial role in various cancers, a richer discussion on the possible pathways and mechanisms through which they influence GC would have added significant depth to the article.

Importantly. the authors have cited potential benefits of certain chemotherapeutic drugs, including dasatinib, AMG 706, imatinib, A-443654 and BIBW2992, solely based on lower estimated IC50 values. This is a very premature and un-insightful claim and may not be relevant in this particular case. Every chemotherapeutic drug has a specific target and mode of action. All the drugs mentioned here are kinase inhibitor, whereas very few of the target proteins mentioned in the manuscript belongs to the kinase family, like PDK4 but not APOD, ACTA2 or SAPCD2. Moreover, each of the drug has a specific target, like dasatinib and imatinib binds to ABL kinase, BIBW2992 binds to EGFR and HER2. This entire section either needs to be re-written with relevant examples or could be left out.

Additional comments

In conclusion, the review serves as a valuable stepping stone in the ongoing exploration of the role of RNA methylation in GC, offering promising avenues for further research. However, to fully capitalize on its potential, it needs to delve deeper, offering a more critical and comprehensive analysis of the existing literature and its own findings, thus paving the way for a richer, more nuanced understanding of the complex processes governing GC development and progression.

---

## Round 0.2 · accepted · Accept

Dear Authors,

Thank you for reviewing your manuscript. The reviewers agree that your work is now suitable for publication.

Thank you for your submission to PeerJ.

Reviewer 1 ·

Basic reporting

The authors have addressed all my concerns.

Experimental design

The authors have addressed all my concerns.

Validity of the findings

No comments.

Reviewer 2 ·

Basic reporting

no comment

Experimental design

no comment

Validity of the findings

no comment

Reviewer 3 ·

Basic reporting

The authors have successfully commented on the concerns raised in the review. The manuscript can be accepted for publication.

Experimental design

No Comment

Validity of the findings

no comment